# COVID-19 Vaccination among HIV+ Patients: An Italian Cross-Sectional Survey

**DOI:** 10.3390/vaccines10091438

**Published:** 2022-09-01

**Authors:** Fabrizio Bert, Alex Pivi, Antonino Russotto, Benedetta Mollero, Gianluca Voglino, Giancarlo Orofino, Roberta Siliquini

**Affiliations:** 1Department of Public Health and Pediatrics Sciences, University of Turin, 10126 Turin, Italy; 2School of Medicine, University of Turin, 10126 Turin, Italy; 3Ministry of Health, 00144 Rome, Italy; 4Amedeo di Savoia Hospital, 10149 Turin, Italy; 5Hospital Molinette, City of Health and Science of Turin, 10126 Turin, Italy

**Keywords:** vaccine, COVID-19, HIV, vaccine hesitancy, risk perception, PLWH

## Abstract

Background: This study aims to evaluate knowledge, attitudes and practices (KAP) of HIV-patients about COVID-19 vaccination. Methods: A cross-sectional study was conducted by administering questionnaires to 160 patients followed by Amedeo di Savoia Hospital in Turin. Statistical analyses were performed in order to identify predictors of infection and severity of COVID-19 disease risk perception. Results: The 86.2% of patients were vaccinated for COVID-19, while 7.6% do not intend to be vaccinated; 50.7% thought that there is a minimal risk to get COVID-19; 85.8% thought that COVID-19 is a serious illness. The 56% and the 36.5 thought that seropositivity carries a greater risk to develop respectively COVID-19-related complications or vaccine complications. At the multivariate analysis having a job, proactive research of vaccine information and being HIV+ for several years are related to a lower risk perception of infection. The perception of COVID-19 severity is influenced by age, by being LGB and by believing that HIV+ status correlates with a higher risk of developing complications from COVID-19. Conclusions: as the pandemic can adversely impact the HIV care with increasing loss to follow-up, vaccination is essential to contrast infection in HIV+ patients. Our findings suggested that some HIV+ patients refuse vaccination against COVID-19.

## 1. Introduction

In December 2019 the first case of Coronavirus (COVID-19) was reported in Wuhan, China. After its spreading at an alarming rate worldwide, the World Health Organization (WHO) described COVID-19 as a Public Health Emergency of International Concern on 30 January 2020, and by 11 March 2020, it was officially declared a pandemic [1]. As of 11 July 2022, over 550 million confirmed cases and over six million deaths have been reported globally while in Italy there have been more than 19 million cases since the beginning of the pandemic, and more than 160,000 deaths [2].

In this emergency context, an unprecedented collaboration between pharmaceutical companies and governments at an international level has led to an acceleration in the vaccine production process [3]. In December 2020, the WHO approved the first vaccine, BNT162b2/COMIRNATY, which contains modified nucleoside mRNA that confers protection against COVID-19 [4]. In Italy, at present, five vaccines have been approved and administered during the vaccination campaign: Pfizer/BioNTech’s Tozinameran [5,6], Moderna’s mRNA-1273 [7,8], AstraZeneca’s Vaxzevria [9,10], Ad26.COV2.S by Janssen (Johnson & Johnson) [11,12] and the last one was Novavax COVID-19 vaccine [13,14]. 

To date, about 90% of the population over 12 and about 35% of the population in the 5–11 age group are vaccinated with the primary cycle. In addition, approximately 88% of eligible subjects either received an additional/booster dose, or did not receive it because they developed the disease after completion of the primary cycle [15].

However, it is necessary to specify how the will of the population to get vaccinated is essential to guarantee the effectiveness of vaccination. In this sense, the acceleration of the production process and the use of technologically innovative vaccines could have made part of the general population suspicious. Other factors that could have influenced the acceptance of vaccination by the population are the perception of the risk of contagion and the severity of the disease, the perception of the safety and efficacy of the vaccine, the attitude of public opinion on vaccination, the pathological and vaccination history, the general practitioner advice, the socio-demographic characteristics of the target population [16,17].

Vaccine hesitancy, even before COVID-19, had been identified as one of the 10 threats to global health by the World Health Organization [18]. Large variability in COVID-19 vaccine hesitancy rates has been reported in different countries and for different target populations. Specifically, the highest COVID-19 vaccine hesitancy rates were recorded in the Middle East, Eastern Europe and Russia [19].

Interestingly, greater vaccine hesitancy was observed in studies including specific target populations, such as patients with chronic diseases of various types [20]. For example, high vaccine hesitancy rates have been identified in the population affected by rheumatic diseases, celiac disease, epilepsy and psychiatric disorders, chronic diseases and older people [21,22,23,24,25]. In the same way, some international studies have analyzed the phenomenon of vaccine hesitancy in the HIV+ population and in various subtypes of the same [26,27,28,29].

Vaccine hesitancy in a category of potentially immunologically fragile subjects, such as People living with HIV (PLWH), certainly deserves attention: as well as in other conditions of vulnerability or immunosuppressive treatment, such as in subjects with Myasthenia Gravis [30], persons living with HIV, could have a higher risk of SARS-CoV-2 infection and mortality risk from COVID-19 than people without HIV [31]. As the pandemic is still ongoing, the PLWH should be aware of the increased risk of COVID-19 infection and mortality, as well as the safety of vaccination in this type of population [32,33]. Furthermore, any isolation due to positivity, as well as potential reductions in therapy and diagnosis due to reduced access to clinics, could have even more damaging effects [34]. 

There are no studies in the literature on COVID-19 vaccine hesitancy in HIV-positive subjects in Italy. For this reason, taking into account the risk of this fragile population, it is essential to understand the compliance to vaccination campaign, the vaccination intention and the risk perception about COVID-19 and its consequences in HIV+ patients. The aim of this study therefore was to evaluate the knowledge, attitudes, and practices (KAP) of HIV-patients about COVID-19 vaccination.

## 2. Materials and Methods

### 2.1. Study Design

A cross-sectional survey was conducted, in the period November 2021–April 2022, involving HIV+ patients followed by the infectious disease clinic of Amedeo di Savoia Hospital in Turin, collecting 160 questionnaires. All of the participants gave their consent to this study and signed an informed consent form after they received information about the finalities and the objectives of the study. The inclusion criteria for the study were: age > 18 yo, HIV diagnosis confirmed by the laboratory, being in visit at the clinic at least for the second time and being able to understand the informed consent and the questionnaire. The authors assert that all procedures contributing to this work had the approval of the Ethics Committee intercompany of Azienda Ospedaliero-Universitaria, City of Health and Science of Turin.

### 2.2. Data Collection

The questionnaires were administered at the infectious disease clinic of Amedeo di Savoia Hospital. The questionnaire consisted of 40 items divided in three sections. In the first section, questions were related to socio-demographic variables, while the second one was centered on patients’ vaccination record, their intention to get vaccinated and their knowledge about COVID-19. In the last part, questions were related to the patients’ personal opinions about the relation between COVID-19 and their seropositivity. Each part of the questionnaire was completed by patients alone with the possibility to ask a researcher for any doubt. All of the information collected during the study was confidential and were treated according the national law about privacy (Legislative Decree n. 196 of 30/6/2003 “Codice in materia di protezione dei dati personali”) and subsequent additions. All data was registered on a single database protected with a password and was analyzed anonymously.

### 2.3. Statistical Analysis

After the data collection, four researchers anonymously filled a single informatic database in the Department of Public Health and Pediatrics at the University of Turin. Descriptive analyses were carried out for all variables. Continuous variables were expressed as mean ± SD, while categorical variables were reported as percentages. The three multivariable logistic regression models were fitted using stepwise forward selection process with a univariable *p*-value less than or equal to 0.25 as main criterion for inclusion: (1) Proactive research of information in the vaccination field, (2) high perception of risk of infection and (3) high perception of Covid severity. The three multivariable logistic regression models were achieved with a stepwise forward selection process, with a univariable *p*-value < 0.250 as the main criterion [35]. Results were expressed as Odds Ratio (OR) and 95% Confidence Interval (95% CI). Stata (v16) was used and a two-tailed *p*-value < 0.05 was considered statistically significant. Missing values were excluded.

## 3. Results

Table 1 provides a summary of the characteristics of 160 participants. The majority (87.1%) of patients had Italian citizenship, 83.1% were male, 60% had a high school diploma or higher; 47% was heterosexual, 37.8% was LGB and 15.2% no longer had sexual intercourse with anyone. The mean of years since HIV diagnosis was 12.03 (Table 1).

In Table 2 we reported information on vaccination status, beliefs about perception of the risk and severity’s infection. In our sample 86.2% of patients were vaccinated for COVID-19, while 7.6% did not intend to be vaccinated. The 50.7% of our sample thought that there is a minimal risk to get COVID-19; the 85.8% thought that COVID-19 is a serious illness, while 14.2% thought that it is not dangerous. In addition, the 56% thought that seropositivity carries a greater risk to develop COVID-19-related complications and 36.5% thought that seropositivity carries a greater risk to develop complications after getting COVID-19 vaccine. In conclusion, 76.1% of our sample declared to have trust in NHS professionals (Table 2).

In addition, we asked patients if they had received any information about vaccination by NHS professionals. The 23.9% of our sample had not received any information. Among those who had received information, 50% were informed at the HIV clinic, 25.7% were informed by their general practitioner, 21.3% by physicians at the vaccination center and the last part of our sample was informed by other specialists. We then analyzed if patients searched for information by themself. In our sample, 61.9% of patients searched for information independently: among these, 40% searched for information on the Internet or through Social Media, 24.4% on television through newspapers or institutional channels, and finally 10.6% through friends. The 7.5% declared to have searched for information by other not specified sources.

In Table 3, by conducting a multivariate analysis and by adjusting it by age and sex was emerged that patients with a degree or a higher title of education searched proactively 3.24 times more than patients who had a diploma. Interestingly, patients who had a higher risk of COVID-19 infection and gravity’s perception had searched less information than those who had a lower risk of infection and gravity‘s perception. Finally, as the years go by from diagnosis, PLWH tend to be less proactive in seeking vaccine information (Table 3).

In Table 4, we conducted a multivariate analysis in order to identify potential predictors of high perception of risk of COVID-19 infection and of high perception of COVID-19 disease severity. It emerged that there were variables related to a lower perception of risk of infection such as having a job, having done a proactive research of information in the vaccination field and being HIV+ for several years.

Regarding the perception of COVID-19 severity, as age increased, the perception of COVID-19 severity also increased. The same applies to being gay, bisexual or lesbian and to the belief that HIV+ status correlates with a higher risk of developing complications from COVID-19 (Table 4).

## 4. Discussion

The main purpose of this study was to evaluate the knowledge, attitudes, and practices (KAP) of HIV-patients about COVID-19 vaccines, in order to identify any variable responsible for possible low vaccination coverage, risk awareness about infection and the severity of COVID-19. Of our participants seventeen were not vaccinated and twelve of them were hesitant to be vaccinated. This indicates that around one out of ten people living with HIV were not vaccinated against SARS-CoV-2 and almost one in ten people were vaccine hesitant, despite having a self-perception that COVID-19 can be a serious disease.

We compared our results with different existing studies in literature among the general Italian population and among PLWH globally. In our sample 10.7% of patients were not vaccinated, this percentage was more than double that one retrieved in another Italian study conducted among the general population in which the percentage of patients who were not vaccinated was around 4.9% [36]. The COVID-19 vaccine unwillingness among our participants was 7.6%. The comparison with other Italian studies in the general population yielded alarming results. Our results, indeed, are relatively higher than findings about general population reported by Zarbo et al. [37], for which vaccine hesitancy was around 6.8%, and even higher than another study for which vaccine hesitancy was around 4% of patients who did not want to get vaccinated [36].

However, in the international context PLWH previously reported vaccine hesitancy rates range from 27.5% to 38.4% across various settings and in particular in China [38,39], France [40], and South India [41]. It is reassuring that the vaccine hesitancy among PLWH in our sample is quite lower than other experiences reported in literature, as the pandemic can adversely impact the HIV care with increasing loss to follow-up or disengagement. Interestingly, we found similar results to a study conducted in developing countries in Latin America: while the percentage of people wanting to get vaccinated or vaccinated and those hesitant are comparable, we found more than three times the percentage of people unwilling to get vaccinated categorically [42].

The proactive search about vaccines was less common among those with a higher perception to be infected with SARS-CoV-2 and in people with a longer story of HIV positivity. The relationship between high perception and proactive search is difficult to explain. A greater number of years since HIV+ diagnosis, instead, can reduce the proactive search about vaccines because patients under treatment reached a quality of life and an attitude to be concerned about their health probably more similar to those of general population. The risk perception to be infected by SARS-CoV-2 is lower in workers, in those who searched information proactively and, again, in patients with a longer story of seropositivity. The proactive search of information about vaccines, in this context, could reduce the risk perception if the patients had found the answers to their doubts on sources like social media and not institutional websites reporting conspiracy and minimizing the pandemics theories. The perception of COVID-19 severity was higher in LGB people, with increasing age and in those that belief that seropositivity expose to a higher risk of developing complications from COVID-19. The reason behind the difference found according to sexual orientation is quite difficult to explain and deserves further studies with greater samples to achieve appropriate answers. The increasing age and the seropositivity status, instead, may have induced an high perception because the communication strategies adopted by mass media and health professionals highlighted the greatest risk of COVID-19 disease and its complications among older and fragile persons.

This study is the first that analyzed knowledge, attitudes, and practices towards COVID-19 vaccination and addressed the phenomenon of vaccine hesitancy among PLWH in Italy. In fact, at the moment there are few studies about COVID-19 vaccine hesitancy in the general Italian population and there are none that specifically regards Italian PLWH. Understanding HIV patients’ perspectives is important to contrast vaccine hesitancy. However, some limitations of our study must be acknowledged. First of all, the cross-sectional design allows only to make hypotheses about association between variables but further research is needed in order to demonstrate cause-effect relationships. Secondly, we recruited an opportunistic sample and this approach may affect the generalisability of our results. However, the sample was composed by recruiting all the eligible patients followed in the study period by the infectious disease clinic of Amedeo di Savoia Hospital in Turin who accepted to participate. In addition, data collection and examination was set to exclude ineligible participants and ensure data quality. In conclusion, we believe that our results can be useful for public health professionals that work with these subgroup of population.

## 5. Conclusions

More studies need to be performed to better analyze the attitudes of PLWH and if they are different from the general population and to better understand factors that are potentially responsible for vaccine hesitancy as it has already been done in other countries for example in the United States of America [43]. In conclusion, COVID-19 is a serious illness that has caused thousands of deaths worldwide and can negatively affect the lives of populations at risk such as HIV+ patients, with potential reductions in therapy, diagnosis and increasing loss to follow-up or disengagement. Vaccination is a tool to contrast the spread and symptoms of this infection in the general population and in HIV+ patients. Our findings suggested that there is a part of HIV+ patients who did not want to get vaccinated against COVID-19 and even a greater percentage who thought that it is not a dangerous disease. Therefore, it is important to spread more awareness among HIV+ patients about the risks of this infection and about vaccines with the aim to contrast vaccine hesitancy.

## Figures and Tables

**Table 1 vaccines-10-01438-t001:** Sample’s description (N = 160).

		N	Mean ± SD
**Age**		158	49.97 ± 11.82
**Years from HIV+ diagnosis**		158	12.03 ± 9.14
		**N**	**%**
**Citizenship**	*Italian*	135	87.1
	*Foreign*	20	12.9
**Gender**	*Male*	133	83.1
	*Female*	27	16.9
**Marital status**	*Single*	99	62.3
	*With a partner*	60	37.7
**Educational leve**	*High school Diploma*	68	42.5
	*Elementary school*	7	4.4
	*Middle school*	57	35.6
	*University Degree or higher*	28	17.5
**Occupation**	*Unemployed*	60	38
	*Worker*	87	55
	*Healthcare worker*	11	7
**Sexual Orientation**	*heterosexual*	71	47
	*LGB*	57	37.8
	*Has not sexual intercourse*	23	15.2
**Unprotected sex**	*No*	109	68.5
	*Yes*	48	30.2
	*Does not respond*	2	1.3
**Drug use**	*No*	149	93.1
	*Yes*	11	6.9
**Cohabitation with HIV+ lpeople**	*No*	138	86.2
	*Yes*	22	13.8
**Chronic conditions**	*No*	111	69.4
	*Yes*	49	30.6
**Lymphocyte count (CD4+, in cells/mm^3^)**	*0–200*	11	7.7
	*201–500*	42	29.6
	*≥501*	89	62.7
**Viral load**	*Undetectable*	119	79.3
	*Present*	31	20.7

**Table 2 vaccines-10-01438-t002:** COVID-19 Data % (n) (N = 160).

		N	%
**Vaccination status**	*Vaccinated*	137	86.2
	*Not vaccinated*	17	10.7
	*Natural immunity*	4	2.5
	*I do not know*	1	0.6
**Vaccination intention**	*Yes*	13	8.2
	*No*	12	7.6
	*Already vaccinated*	126	79.8
	*I do not know*	7	4.4
**Perception of infection’s risk**	*Low*	77	50.7
	*High*	75	49.3
**Perception of COVID-19 severity**	*Low*	22	14.2
	*High*	133	85.8
**Belief that seropositivity is related to an higher risk of developing complications from COVID-19**	*No*	75	56
	*Yes*	57	44
**Belief that seropositivity is related to an higher risk of developing complications after COVID-19 vaccination**	*No*	50	36.5
	*Yes*	87	63.5

**Table 3 vaccines-10-01438-t003:** Proactive search of information in the vaccination field (multivariate analysis) (N = 160) *.

		Univariate OR (CI95%)	*p*-Value	Adjusted OR (CI95%)	*p*-Value
**Educational level**	*High school Diploma*	1	-	1	-
	*Elementary school*	0.93 (0.19–4.50)	0.931	1.10 (0.11–10.71)	0.937
	*Middle school*	0.90 (0.44–1.83)	0.762	1.37 (0.52–3.59)	0.521
	*Degree or higher*	3.22 (1.09–9.49)	**0.034**	3.24 (0.93–11.36)	0.066
**Perception of the risk of COVID-19 infection**	*Low*	1	-	1	-
	*High*	0.48 (0.25–0.94)	**0.033**	0.33 (0.13–0.82)	**0.017**
**COVID-19 severity of disease perception**	*Low*	1	-	1	-
	*High*	0.33 (0.11–1.05)	0.060	0.27 (0.07–1.11)	0.070
**Greater risk of COVID-19 complications in HIV+**	*No*	1	-	1	-
	*Yes*	0.59 (0.29–1.22)	0.150	0.97 (0.39–2.45)	0.954
**Years since HIV diagnosis**	*Years HIV+*	0.96 (0.39–0.99)	**0.022**	0.94 (0.89–0.99)	**0.027**

* Adjusted by age and gender.

**Table 4 vaccines-10-01438-t004:** Potential predictors of high perception of risk of infection and high perception of COVID severity (N = 160).

		High Perception of Risk of Infection adjOR (IC95%)	*p*-Value	High Perception of COVID-19 Severity adjOR (IC95%)	*p*-Value
**Nationality**	*Italian*	1	-	1	-
	*Foreign*	0.13	0.150	1	0.999
**Age**		0.99	0.746	1.22	**0.026**
**Gender**	*Male*	1	-	1	-
	*Female*	4.75	0.111	56.55	0.138
**Marital status**	*Single*	1	-	1	-
	*With a partner*	0.63	0.430	4.55	0.177
**Educational level**	*Diploma high school*	1	-	1	-
	*Elementary school*	1	0.999	0.25	0.522
	*Middle school*	0.66	0.609	70.38	0.092
	*Degree or higher*	1.31	0.726	0.77	0.845
**Occupation**	*Unemployed*	1	-	1	-
	*Worker*	0.11	**0.016**	3.16	0.518
	*Health worker*	0.06	0.055	5.65	0.574
**Sexual Orientation**	*Heterosexual*	1	-	1	-
	*LGB*	3.01	0.176	155.40	**0.031**
	*No sexual intercourses*	0.95	0.954	0.58	0.713
**Unprotected sex**	*No*	1	-	1	-
	*Yes*	1.97	0.325	1.48	0.663
**Har received information about vaccines**	*No*	1	-	1	-
	*Yes*	1.26	0.702	15.56	0.055
**Proactive search of information about vaccines**	*No*	1	-	1	-
	*Yes*	0.13	**0.003**	0.12	0.164
**Years from diagnosis**		0.93	**0.042**	0.87	0.126
**Belief that seropositivity is related to an higher risk of developing complications from COVID-19**	*No*	1	-	1	-
	*Yes*	1.42	0.587	131.13	**0.013**
**Belief that seropositivity is related to an higher risk of developing complications after COVID-19 vaccination**	*No*	1	-	1	-
	*Yes*	2.24	0.280	0.79	0.889
**Lymphocyte count**	0–200	1	-	1	-
	201–500	2.09	0.309	0.65	0.715
	≥501	4.94	0.293	0.74	0.896
**Viral load**	*Undetectable*	1	-	1	-
	*Detectable*	1.28	0.723	0.86	0.885

## Data Availability

The data that support the findings of this study are available from the corresponding author, A.R., upon reasonable request.

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
