# Peer review of "COVID-19 Vaccination among HIV+ Patients: An Italian Cross-Sectional Survey"

_vaccines, 2022, doi:10.3390/vaccines10091438_

Round 1

Reviewer 1 Report

This study explores the relevant topic of COVID-19 vaccines’ attitude and hesitancy among HIV+. The paper sounds interesting, quite organized and comprehensive. The design of the study is good. The results are in line with the discussion and methods. Vaccination against COVID-19 has raised many concerns in public opinion and studies in special fragile populations are needed. I think that it is a very relevant topic that must be addressed. I only have some suggestions:

-Introduction/background (line 71-72 and 74-75). What about patients affected by autoimmune disease who are treated with immunosuppressants? (i.e., Myasthenia gravis, IBD, transplant receivers, etc.). This topic is fully discussed in a very recent study exploring the impact of COVID in Myasthenia Gravis and safety of vaccines in these fragile patients (https://doi.org/10.3390/ neurolint14020033). These results underlined the fact that unvaccinated patients often present high mortality rates from COVID infections compared to vaccinates patients. I suggest to report these recent data and add it to the introduction briefly.

-lines 78-80: Were there any interaction between antivirals and vaccination explored? What about the safety of vaccines in HIV patients? The safety in this population should be discussed and clearly confirmed when appropriate.

-Table 1. “Unprotected sex”: this concept should be widened; indeed, sexual protection is not enough for prevention of COVID. What about oral (and, of course, respiratory) contacts? This topic is very practical and relevant to limit spreading contagions.

-lines 150-151: percentages are not clearly assigned to categories. Social media 40%? The sentence is ambiguous, please adjust it.

-lines 181-182: How was the prognosis in unvaccinated patients?

-lines 198-199: the loss at follow-up is a relevant problem for chronic disease. I suggest to add this consideration in the abstract and conclusions.

-there are no relevant grammar mistakes. Line 131 there is a missing “%”.

Author Response

Point 1: Introduction/background (line 71-72 and 74-75). What about patients affected by autoimmune disease who are treated with immunosuppressants? (i.e., Myasthenia gravis, IBD, transplant receivers, etc.). This topic is fully discussed in a very recent study exploring the impact of COVID in Myasthenia Gravis and safety of vaccines in these fragile patients (https://doi.org/10.3390/ neurolint14020033). These results underlined the fact that unvaccinated patients often present high mortality rates from COVID infections compared to vaccinates patients. I suggest to report these recent data and add it to the introduction briefly.

Response 1: we accepted the suggestion.

Point 2: lines 78-80: Were there any interaction between antivirals and vaccination explored? What about the safety of vaccines in HIV patients? The safety in this population should be discussed and clearly confirmed when appropriate.

Response 2: we accepted suggestion and inserted two references.

Point 3: Table 1. “Unprotected sex”: this concept should be widened; indeed, sexual protection is not enough for prevention of COVID. What about oral (and, of course, respiratory) contacts? This topic is very practical and relevant to limit spreading contagions.

Response 3: unfortunately we have not investigated risk behaviors for the infection transmission.

Point 4: lines 150-151: percentages are not clearly assigned to categories. Social media 40%? The sentence is ambiguous, please adjust it.

Response 4: we corrected the sentence as required.

Point 5: lines 181-182: How was the prognosis in unvaccinated patients?

Response 5: unfortunately we have not investigated the prognosis of unvaccinated patients given the small number of them.

Point 6: lines 198-199: the loss at follow-up is a relevant problem for chronic disease. I suggest to add this consideration in the abstract and conclusions.

Response 6: we accept the suggestion and inserted a small sentence both in the abstract and in the conclusions.

Reviewer 2 Report

Introduction

In the same way, some international studies have analyzed the phenomenon of vaccine hesitancy in the HIV+ population and in various subtypes of the same 76 [26–29].

[please provide % details as this is key to your argument]

Sample

“being in visit at the clinic at least for the second time”

[please explain why did you not include those with one visit?]

[What is the total number of enrollees in the clinic? If some had not attended within your time frame, please explain why you did not conduct phone questionnaires?]

[What is not clear is the total number of eligibles.]

[How do these exclusions affect your generalisability?]

Methods

“The questionnaire consisted of 40 items divided in three sections. In the first section, questions were related to socio-demographic variables, while the second one was centered on patients’ vaccination record, their intention to get vaccinated and their knowledge about Covid-19. In the last part, questions were related to the patients’ personal opinions about the relation between Covid-19 and their seropositivity.”

[are there any data on reliability from previous uses of questions or the questionnaire please?]

Results

“15.23% had not sexual intercourse with anyone.” [how did they contract HIV? Contaminated blood products or IV injection?

“we asked patients if they had received any information about vaccination by NHS professionals. The 23.90% of our sample had not received any information. Among those who had received information, 50% were informed at the HIV clinic, 25.65 were informed by their general practitioner, 21.25% by physicians at the vaccination center and the last part of our sample was informed by other specialists”

[do you know what information they received please?]

[Please provide the numbers of subjects for the different lines in the tables and consider if the numbers are large enough to merit statistical analysis]

The article is overlong for a small sample like this (agreed it’s the only study in Italy you found) so please consider reducing the word count].

Author Response

Point 1: “In the same way, some international studies have analyzed the phenomenon of vaccine hesitancy in the HIV+ population and in various subtypes of the same 76 [26–29].” [please provide % details as this is key to your argument]

Response 1: We did not provide % details in the introduction because we preferred to discuss and analyze the most interesting articles in the discussion. 

Point 2: Sample, “being in visit at the clinic at least for the second time” [please explain why did you not include those with one visit?] [What is the total number of enrollees in the clinic? If some had not attended within your time frame, please explain why you did not conduct phone questionnaires?] [What is not clear is the total number of eligibles.] [How do these exclusions affect your generalisability?]

Response 2: We did not include patients who came at the clinic for the first time for practical reasons in order to have complete data. In fact, patients who came at least for the second time had already lymphocyte count (CD4+, in cells/mm3) and viral load online. The total number of eligibles estimated in the study period was around 450 persons. We decided to not conduct phone questionnaires since we wanted to guarantee the same conditions of participation to all our subjects. The opportunistic approach may affect the generalisability of our results. However, we recruited all the HIV+ individuals attending to the main clinic for HIV management of Piedmont who accepted to participate and we believe that our results can be useful for public health professionals that work with these subgroup of population.

Point 3: “The questionnaire consisted of 40 items divided in three sections. In the first section, questions were related to socio-demographic variables, while the second one was centered on patients’ vaccination record, their intention to get vaccinated and their knowledge about Covid-19. In the last part, questions were related to the patients’ personal opinions about the relation between Covid-19 and their seropositivity.” [are there any data on reliability from previous uses of questions or the questionnaire please?]

Response 3: We thank the referee for this important comment. We analyzed the scientific literature and we selected the questions according to similar studies previously conducted in similar settings. We proposed, then, the questionnaire through a pilot study to 20 HIV+ people in order to investigate his understandability before to recruit the final sample.

Point 4: “15.23% had not sexual intercourse with anyone.” [how did they contract HIV? Contaminated blood products or IV injection?

Response 4: We rephrased the sentence.

Point 5: “we asked patients if they had received any information about vaccination by NHS professionals. The 23.90% of our sample had not received any information. Among those who had received information, 50% were informed at the HIV clinic, 25.65 were informed by their general practitioner, 21.25% by physicians at the vaccination center and the last part of our sample was informed by other specialists” [do you know what information they received please?]

Response 5: Unfortunately, we did not dedicate a section of the questionnaire to study the information received from colleagues.

Point 6: [Please provide the numbers of subjects for the different lines in the tables and consider if the numbers are large enough to merit statistical analysis]

Response 6: We added the numbers of subjects in table 4. We discussed our sample's size and considering that PLWH is a very specific population we think that it merit a statistical analysis.

Point 7: The article is overlong for a small sample like this (agreed it’s the only study in Italy you found) so please consider reducing the word count].

Response 7:  We rephrased one period in the introduction to reduce word count by 100. 

Reviewer 3 Report

The manuscript describes knowledge, attitudes and practices towards COVID-19 vaccination in PLWH, a group of subjects exposed at greater risk of infection and critical illness. The authors analyzed data from 160 outpatient subjects, who exhibit a worrisome level of willingness to get the vaccine and wrong disease perception.  

In this sense, the paper is timely and thought-provoking: it can be acceptable for publication in Vaccines after minor revision.

- Authors might want to add more relevant keywords, such as vaccine hesitancy, risk perception, PLWH.

- Lines 123-124 can be changed to “The three multivariable logistic regression models were fitted using stepwise forward selection process with a univariable p-value less than or equal to 0.25 as main criterion for inclusion”. 

- Table 1: In Educational level, does “Degree or higher” refer to University degree? 

- Table 1: In Viral load, change to “Present”

- Table 1: Lymphocyte count could be rewritten as “Lymphocyte count (CD4+, in cells/mm3)

Author Response

Point 1: Authors might want to add more relevant keywords, such as vaccine hesitancy, risk perception, PLWH.

Response 1: We accept the suggestion.

Point 2: Lines 123-124 can be changed to “The three multivariable logistic regression models were fitted using stepwise forward selection process with a univariable p-value less than or equal to 0.25 as main criterion for inclusion”. 

Response 2: We accept the suggestion.

Point 3: Table 1: In Educational level, does “Degree or higher” refer to University degree?

Response 3: We clarified the level education.  

Point 4: Table 1: In Viral load, change to “Present”

Response 4: We corrected the misspelling.

Point 5: Table 1: Lymphocyte count could be rewritten as “Lymphocyte count (CD4+, in cells/mm3)

Response 5: We accept the suggestion. 

Round 2

Reviewer 2 Report

The authors have made some small changes and importantly described their sample as opportunistic. 

I hadn't noticed it before but the % results to two decimal places are inappropriately precise for a small opportunistic sample of 160. Please change e.g. 56.43% to 56.4%

Author Response

Point 1: I hadn't noticed it before but the % results to two decimal places are inappropriately precise for a small opportunistic sample of 160. Please change e.g. 56.43% to 56.4%

Response 1: we approximated the % results as required throughout the article including the tables.